



**Organic enrichment in droplet residual particles relative to out of cloud over the northwest**
**Atlantic: Analysis of airborne ACTIVATE data**
Hossein Dadashazar[1], Andrea F. Corral[1], Ewan Crosbie[2,3], Sanja Dmitrovic[4], Simon Kirschler[5,6],
Kayla McCauley[7], Richard Moore[2], Claire Robinson[2,3], Joseph Schlosser[1], Michael Shook[2], K.
Lee Thornhill[2], Christiane Voigt[5,6], Edward Winstead[2,3], Luke Ziemba[2], Armin Sorooshian[1,4,7]
[1]Department of Chemical and Environmental Engineering, University of Arizona, Tucson, AZ,
USA
[2]NASA Langley Research Center, Hampton, VA, USA
[3]Science Systems and Applications, Inc., Hampton, VA, USA
[4]James C. Wyant College of Optical Sciences, University of Arizona, Tucson, AZ, USA
[5]Institute of Atmospheric Physics, German Aerospace Center
[6]Institute of Atmospheric Physics, University Mainz, Germany
[7]Department of Hydrology and Atmospheric Sciences, University of Arizona, Tucson, AZ, USA

[*]Correspondence to: Hossein Dadashazar (hosseind@arizona.edu)



**Abstract**.

Cloud processing is known to generate aerosol species such as sulfate and secondary organic aerosol, yet there is a scarcity of airborne data to examine this issue. The NASA Aerosol Cloud meTeorology Interactions oVer the western ATlantic Experiment (ACTIVATE) was designed to build an unprecedented dataset relevant to aerosol-cloud interactions with two coordinated aircraft over the northwest Atlantic, with aerosol mass spectrometer data used from four deployments between 2020-2021 to contrast aerosol composition below, in (using a counterflow virtual impactor), and above boundary layer clouds. Consistent features in all time periods of the deployments (January-March, May-June, August-September) include the mass fraction of organics and relative amount of oxygenated organics (m/z 44) relative to total organics ($f_{44}$) increasing in droplet residuals relative to below and above cloud. Detailed analysis comparing data below and in cloud suggests a possible role for in-cloud aqueous processing in explaining such results. These results are important as other datasets (e.g., reanalysis) suggest that sulfate is both more abundant than organics (in contrast to this work) and more closely related to drop number concentrations in the winter when aerosol-cloud interactions are strongest; here we show that organics are more abundant than sulfate in the droplet residuals and that aerosol interaction with clouds potentially decreases particle hygroscopicity due to the significant jump in organic:sulfate ratio for droplet residuals relative to surrounding cloud-free air. These results are important in light of the growing importance of organics over the northwest Atlantic in recent decades relative to sulfate owing to the success of regulatory activity over the eastern United States to cut sulfur dioxide emissions.





## 1. Introduction


46       The nature of aerosol-cloud interactions over the northwest Atlantic Ocean is uncertain
even though the region has been the target of decades of atmospheric research (Sorooshian et al.,
2020). These interactions include a subset of aerosol particles called cloud condensation nuclei
(CCN) that activate into cloud droplets, which subsequently undergo aqueous processing to
transform into a particle after evaporation varying in size and composition relative to the original
CCN. An aspect of these steps that is poorly characterized is the composition of the droplet
residuals in cloud relative to particles below and above clouds, which requires airborne
measurements. The NASA Aerosol Cloud meTeorology Interactions oVer the western ATlantic
Experiment (ACTIVATE) was designed to collect in situ and remote sensing data in and around
clouds during different seasons in a region with a wide range of weather conditions (Painemal et
al., 2021) and air mass sources (Corral et al., 2021), qualifying as a suitable dataset to examine
this very issue.

58       The annual cycle of aerosol and cloud drop number concentrations ($N_d$) varies in the
northwest Atlantic, with aerosol parameters (e.g., aerosol optical depth, aerosol index) peaking in
summer months in contrast to $N_d$ being highest the winter (Figure S1). This discrepancy was
reconciled by Dadashazar et al. (2021a) who showed that conditions linked to cold air outbreak
events (e.g., enhanced turbulence, marine boundary layer height, low-level liquid cloud fraction)
promote stronger aerosol-cloud interactions in the winter to help activate particles into drops with
higher efficiency than other times of the year. Gradient boosted regression tree analysis revealed
that the most influential aerosol parameter in predicting $N_d$ was either surface mass concentration
of sulfate (winter) or organics (summer). However, those results were based on reanalysis data
without any indication of causal effects between aerosol composition and cloud microphysics.
Airborne in situ data are needed to unravel the composition details in and around clouds. Or
particular interest related to aerosol chemical characterization around clouds is growing evidence
in the literature that in-cloud aqueous processing can generate not only sulfate (Barth et al., 2000;
Ervens, 2015) but also secondary organic aerosol (SOA) (Blando and Turpin, 2000; Warneck,
2003; Sorooshian et al., 2006a; Ervens et al., 2011; Heald et al., 2011), which is hypothesized to
manifest itself in enhanced organic mass fractions in droplet residuals relative to below and above
cloud. Past work over the northwest Atlantic has pointed to the importance of secondary formation
via gas-to-particle conversion processes in influencing the organic carbon budget of aerosol
particles (de Gouw et al., 2005; Schroder et al., 2018; Shah et al., 2019). Furthermore, chemical
analysis of droplet residuals can lend insight into properties of the CCN activating into droplets,
with past work showing an important role for organics (Russell et al., 2000; Drewnick et al., 2007;
Mertes et al., 2007; Hawkins et al., 2008; Asa-Awuku et al., 2015).

80       The goal of this study is to compare aerosol mass spectrometer data over the northwest
Atlantic below, in, and above clouds for different times of the year (February-March, May-June,
August-September). Case studies of flights during cold air outbreaks probe deeper to better
understand the nature of aerosol and droplet residual particle composition during these events with
stronger aerosol-cloud interactions as compared to other times of the year (Dadashazar et al.,
2021a; Painemal et al., 2021). The results have implications for aerosol-cloud interactions as
droplet residual composition is shown here to deviate from that of aerosol out of cloud. This is
important to lend insight into properties of the CCN activating into drops and/or pointing to a key
role for cloud processing over the northwest Atlantic to alter aerosol properties.






## 2. Methods

### 2.1 Field Campaign Description

We use airborne in situ data collected aboard the HU-25 Falcon from deployments 1 (14 February – 12 March 2020), 2 (13 August – 30 September 2020), 3 (27 January – 2 April 2021), and 4 (13 May – 30 June 2021) of the ACTIVATE mission. Data necessary for this study were only available for two flights in deployment 3 (29 January and 3 February) owing to an aircraft maintenance issue reducing the size of the available payload. ACTIVATE employs a dual aircraft approach with the Falcon acquiring in situ data for trace gases, aerosol particles, and clouds in the marine boundary layer while a King Air flies overhead at ~9 km conducting remote sensing measurements and launching dropsondes (Sorooshian et al., 2019). Typical flights are ~3-4 hours based out of NASA Langley Research Center in Hampton, Virginia. The Falcon flies in what are termed "ensembles", which comprise legs in the following nominal order: below cloud base (BCB), above cloud base (ACB), BCB, ACB, minimum altitude leg at ~150 m (Min. Alt.), above cloud top (ACT), below cloud top (BCT), and then descent back to BCB to start a new ensemble. Cloud-free ensembles include the following legs: Min. Alt., below boundary layer top (BBL), above boundary layer top (ABL), and then descent back down to Min. Alt. to start a new ensemble. The Falcon flies at ~120 m s$^{-1}$, with the duration (length) of each leg and ensemble being ~3.3 min (~24 km) and 35 min (~250 km), respectively. The repeated nature of these ensembles has built a large statistical database relevant to aerosol-cloud-meteorology interactions. Locations of clear and cloudy ensembles are shown in Figure S2, with clear ensembles generally closer to the coast.

### 2.2 Airborne Instrument Details

The central dataset relevant to aerosol composition in this study comes from the Aerodyne High-Resolution Time-of-Flight Aerosol Mass Spectrometer (AMS) (DeCarlo et al., 2008). The instrument measures submicrometer non-refractory aerosol composition in 1 Hz Fast-MS mode with data averaged to 25-second time resolution. We make use of specific mass spectral markers including m/z 43 (mostly $C_2H_3O^+$) and 44 ($CO_2^+$), which represent oxygenated organic fragments, with the ratios of the markers relative to total organic mass referred to as $f_{43}$ and $f_{44}$, respectively. AMS measurements were conducted downstream of an isokinetic double diffuser inlet (Brechtel Manufacturing Inc.) in cloud-free conditions and downstream of a counterflow virtual impactor (CVI) inlet (Brechtel Manufacturing Inc.) in clouds (Shingler et al., 2012). For classification of data as cloud and cloud-free, we use a liquid water content (LWC) threshold of 0.05 g m$^{-3}$ based on data from the Fast Cloud Droplet Probe (FCDP; $D_p$ ~3 – 50 µm) (SPEC Inc.; Kirschler et al., 2022). This LWC threshold has been used in recent work using ACTIVATE data (Dadashazar et al., 2021a). We also use a proxy for hygroscopicity in the form of f(RH), which is the ratio of total light scattering between relative humidities of 80% and 20% as measured by tandem nephelometers (TSI Inc, St. Paul, MN, USA; Model 3563) (Ziemba et al., 2013).

Note that while cloud water samples were also chemically characterized, those data are outside the scope of this work (i) to maintain consistency in AMS data for out-of-cloud and in-cloud data, and (ii) because the total organic fraction could not be quantified owing to only being able to speciate selected organic acids. Furthermore, particle-into-liquid sampler data are not used





owing to lengthier time resolution (~5 min) and innate chemical smearing (Sorooshian et al.,
2006b) preventing a clear assignment of data to individual legs in ensembles.
**2.3 Complementary Datasets**
**2.3.1 HYSPLIT and CWT Maps**
We obtained 5-day back-trajectory data from NOAA's Hybrid Single-Particle Lagrangian
Integrated Trajectory (HYSPLIT) model (Stein et al., 2015; Rolph, 2017) ending at the Falcon
position during any of the 29,164 cloud-free AMS data points (Figure S3). We relied on the
National Centers for Environmental Prediction/National Center for Atmospheric Research
(NCEP/NCAR) reanalysis data using the "Model vertical velocity" method and obtained data
points every 6 hours along trajectories. Altitude histories of the trajectories for each season are
shown in Figure S4.
As this study is mainly focused on sulfate and organics, concentration-weighted trajectory
(CWT) maps were generated using HYSPLIT back-trajectories in conjunction with speciated AMS
data to show the predominant sources for each of these two aerosol components (e.g., Hsu et al.,
2003). As demonstrated by past works for other regions (e.g., Dadashazar et al., 2019), the method
assigns a weighted concentration to grid cells based on mean concentrations passing through each
grid cell from all the considered trajectories. CWT profile maps are produced using the GIS-based
software called TrajStat (Wang et al., 2009).
**2.3.2 MERRA-2**
We use both total and speciated (sulfate and organic) aerosol optical depth (AOD) at 550
nm from the Modern-Era Retrospective analysis for Research and Applications-Version 2
(MERRA-2) (Gelaro et al., 2017) between January 2013 and December 2017 near Aqua's overpass
time (13:30 local time). We also show results for aerosol index (AI), which is the product of AOD
and the Ångström parameter. As the latter accounts for aerosol size, AI is better related to columnar
CCN as compared to AOD (Nakajima et al., 2001). Data are used for the spatial area over the
northwest Atlantic where ACTIVATE data were collected (boxes 1-3 in Figure 1).
**2.3.3 CERES-MODIS**
Cloud droplet number concentrations ($N_d$) are presented for the ACTIVATE region
following the specific calculations and filtering methods of Dadashazar et al. (2021a) using Clouds
and the Earth's Radiant Energy System (CERES) edition 4 products (Minnis et al., 2011; Minnis
et al., 2021). CERES retrieval algorithms are applied to MODerate resolution Imaging
Spectroradiometer (MODIS)-Aqua radiances as obtained during daytime overpasses around 13:30
local time. Level 3 cloud data were used between January 2013 and December 2017 at $1° \times 1°$
resolution for low-level clouds (> 700 hPa) based on CERES-MODIS edition 4 Single Scanning
Footprint (SSF) products (Loeb et al., 2016). $N_d$ was calculated with an adiabatic cloud model
(Grosvenor et al., 2018):





$$N_d = \frac{\sqrt{5}}{2\,\pi\,k}\left(\frac{f_{ad}\,C_w\,\tau}{Q_{ext}\,\rho_w\,r_e{}^5}\right)^{1/2} \qquad\qquad\qquad (1)$$

where k represents the droplet spectrum width (assumed to be 0.8 over the ocean), $r_e$ is cloud drop
effective radius, $\tau$ is cloud optical depth, $Q_{ext}$ is the unitless extinction efficiency factor (assumed
to be 2 for liquid droplets), and $\rho_w$ is the density of water (1 g cm$^{-3}$). $N_d$ data are used when low-
level liquid cloud fraction exceeded 40%. Data are used for the same spatial area as MERRA-2
data (i.e., boxes 1-3 in Figure 1).

**2.4 Classification of Cold Air Outbreak flights**
We determine whether flights occurred during cold air outbreaks (CAOs) leveraging
methods in recent ACTIVATE studies (Seethala et al., 2021; Corral et al., 2022). Briefly, Visible
Infrared Imaging Radiometer Suite (VIIRS) imagery (NASA Worldview) is used to visually
identify cloud streets that are characteristic of CAOs. Flight notes and weather forecast slides were
used as additional confirmation, followed by data from dropsondes released from the King Air
following the method described in Papritz et al. (2015).

**3. Results**
**3.1 Multi-season overview of AMS composition**
Relative to all AMS species, sulfate and organics are the dominant aerosol components by
mass with combined mass fractions being near 75% usually regardless of season or location
relative to clouds (Table 1; spatial maps in Figure 1); this is consistent with their predictive
capability for $N_d$ over the northwest Atlantic (Dadashazar et al., 2021a). Nitrate and ammonium
were the next most abundant components, with chloride being much lower. The highest organic
concentrations were in August-September assisted in part by transported wildfire emissions from
western North America (Mardi et al., 2021). Mean vertical profiles of organics in each season
(Figure S5) show that in all months, but especially May-June and August-September, there is an
enhancement at altitudes exceeding 200 m in the northernmost parts of the study region. Organic
aerosol CWT maps reveal significant influence from continental sources based on the highest
concentrations along trajectories coming from the U.S. East Coast (Figure 2). In terms of the nature
of the organic aerosol fraction, vertical profiles of $f_{44}$ were fairly similar between seasons and areas
of the study region (Figure S5), ranging in mean value for the various leg types in Table 1 between
0.11 and 0.27. For reference, the $f_{44}$ of atomized oxalic acid, a tracer for cloud processing in the
absence of biomass burning and coarse aerosol (Hilario et al., 2021 and references therein), is 0.36
(Lambe et al., 2011).
In contrast to organics, sulfate exhibits more spatially homogenous concentrations over the
northwest Atlantic (Figure 1) owing largely to ocean-emitted dimethylsulfide that undergoes gas
and in-cloud oxidation such as what was shown for the eastern North Atlantic (Ovadnevaite et al.,
2014). This is supported by how sulfate's seasonal CWT maps (Figure 3) differ from those of





organics with comparable concentrations widespread over the northwest Atlantic relative to the
continent. The August-September CWT map for sulfate reveals more high concentration areas
(note the different color bar scale for Aug-Sep in Figure 3) over the continent with concentrations
exceeding those over most of the ocean; this is presumably due to more secondary formation
stemming from local sulfur dioxide emissions over the eastern U.S. (Yang et al., 2018) aided in
part by higher temperatures and humidity (Corral et al., 2021) that co-vary with other conditions
favorable for sulfate production such as stagnation and certain air flow patterns (Tai et al., 2010).
Figure S5 demonstrates that neither sulfate or organics exhibit a clear reduction with altitude
pointing towards a potential source aloft include long-range transport and/or secondary production.
Although based on only two consecutive days of flight data, results from Leaitch et al.
(2010) are relevant in that they sampled below, in, and above boundary clouds over the northwest
Atlantic. On the first day with more marine influence, sulfate was more abundant than organics in
fine particles below cloud. In contrast, the second day had more continental influence with organic
levels exceeding those of sulfate below cloud, which was often the case during ACTIVATE (Table
1). They concluded with a parcel model that the impact of anthropogenic carbonaceous
components on the cloud albedo effect can exceed that of anthropogenic sulfate, which motivates
attention to the droplet residual composition, which is discussed next.
**Table 1. Average concentrations of submicrometer aerosol species measured by an airborne**
**AMS for different seasons associated with ACTIVATE deployments 1-4. Non-CAO and**
**CAO categories include samples collected between January and March. CVI = droplet**
**residual particle measurements in cloud; BCB = below cloud base, ACT = above cloud top,**
**BBL = below boundary layer top, ABL = above boundary layer top. Corresponding standard**
**deviations and number of points are provided in Table S1.**

| | (Non-CAO/CAO/May-Jun/Aug-Sep) | | | |
| --- | --- | --- | --- | --- |
| | CVI | BCB | ACT | BBL | ABL |
| Organic (µg m⁻³) | - | 1.07/0.67/1.49/3.27 | 0.61/0.19/2.62/3.04 | 2.59/1.16/3.49/4.46 | 0.94/0.57/5.28/5.57 |
| Sulfate (µg m⁻³) | - | 0.93/0.79/1.71/1.35 | 0.53/0.26/1.23/1.11 | 0.80/0.57/1.17/1.77 | 0.51/0.45/1.26/2.13 |
| Nitrate (µg m⁻³) | - | 0.40/0.21/0.07/0.16 | 0.19/0.05/0.14/0.11 | 0.79/0.93/0.17/0.21 | 0.14/0.32/0.26/0.19 |
| Ammonium (µg m⁻³) | - | 0.45/0.32/0.36/0.36 | 0.28/0.10/0.41/0.37 | 0.67/0.65/0.38/0.53 | 0.26/0.30/0.51/0.63 |
| Chloride (µg m⁻³) | - | 0.03/0.02/0.03/0.03 | 0.02/0.01/0.02/0.02 | 0.05/0.01/0.02/0.02 | 0.01/0.01/0.02/0.02 |
| Organic$_{MF}$ | 0.55/0.60/0.68/0.61 | 0.40/0.34/0.35/0.48 | 0.28/0.29/0.42/0.51 | 0.50/0.39/0.63/0.57 | 0.44/0.32/0.65/0.54 |
| Sulfate$_{MF}$ | 0.24/0.19/0.14/0.14 | 0.39/0.45/0.53/0.39 | 0.42/0.46/0.43/0.34 | 0.24/0.20/0.26/0.33 | 0.35/0.36/0.24/0.35 |
| Nitrate$_{MF}$ | 0.05/0.05/0.05/0.05 | 0.08/0.07/0.02/0.03 | 0.08/0.07/0.03/0.03 | 0.11/0.22/0.03/0.03 | 0.06/0.14/0.03/0.03 |
| Ammonium$_{MF}$ | 0.09/0.08/0.07/0.09 | 0.13/0.13/0.10/0.08 | 0.20/0.16/0.12/0.10 | 0.14/0.18/0.08/0.07 | 0.14/0.16/0.07/0.08 |
| Chloride$_{MF}$ | 0.06/0.08/0.06/0.10 | 0.01/0.01/0.01/0.01 | 0.01/0.02/0.01/0.01 | 0.01/0.01/0.01/0.00 | 0.01/0.03/0.00/0.00 |
| $f_{44}$ | 0.33/0.34/0.24/0.37 | 0.15/0.13/0.11/0.14 | 0.26/0.16/0.12/0.15 | 0.16/0.14/0.12/0.14 | 0.17/0.14/0.11/0.14 |




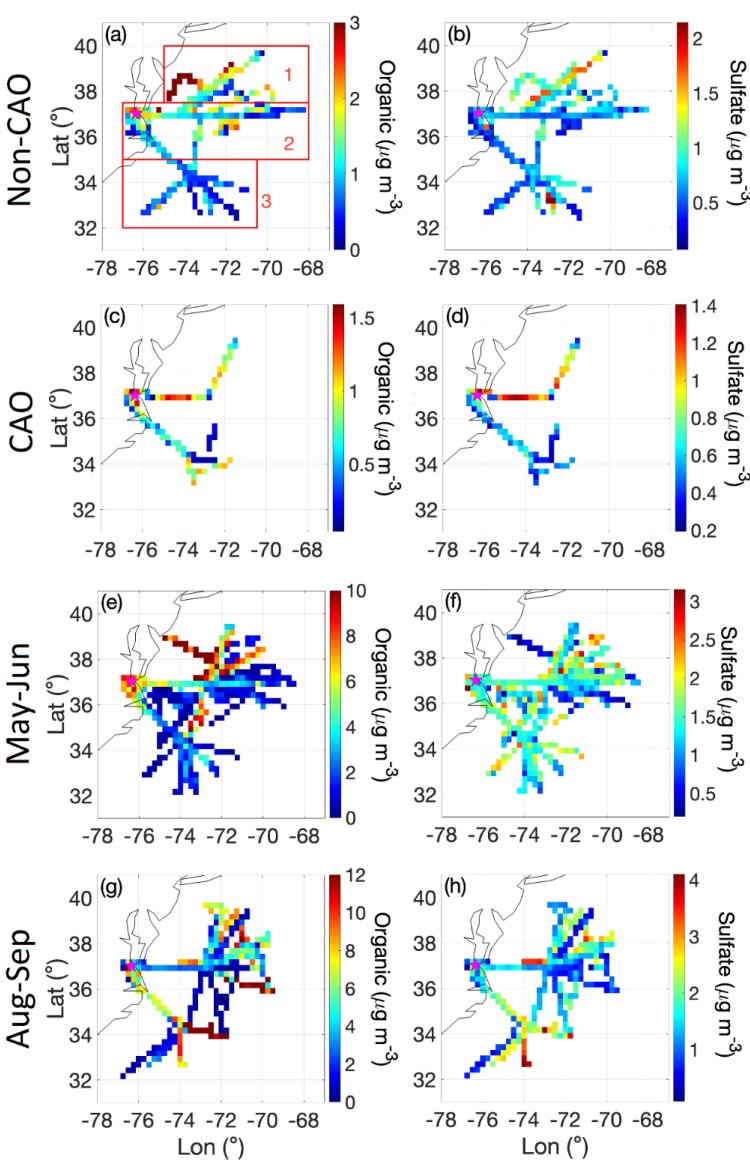

**Figure 1. Spatial map of cloud-free AMS data for organics and sulfate collected during deployments 1-4 of ACTIVATE spanning from February 2020 to June 2021. Non-CAO and CAO represent non cold air outbreak and cold air outbreak days between January and March. Spatial boxes labeled 1-3 in (a) correspond to domains used for calculations in other parts of this study. Grid cells are $0.25° \times 0.25°$ and represent an average of data across all vertical levels flown between 0.02 and 8.1 km. Color bar scales differ by panel to highlight variability better within a panel.**



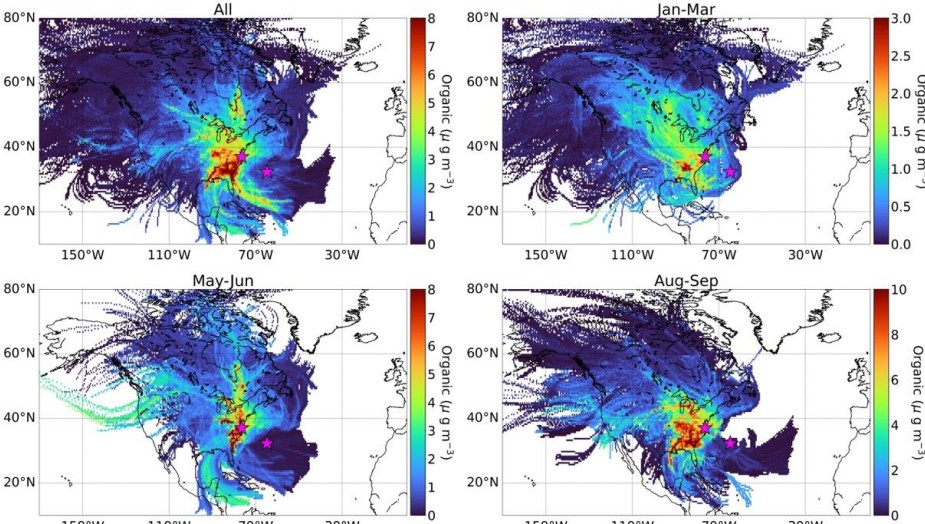

239

**Figure 2. Concentration weighted trajectory maps for organic aerosol concentrations as measured by an AMS on the Falcon during different ACTIVATE deployments (All data, Jan-Mar 2020 and 2021, May-Jun 2021, August-September 2020). These are based on 29,164 cloud-free AMS data points. The pink stars represent NASA Langley Research Center (Hampton, Virginia) and Bermuda for reference. Color bar scales differ to show variability better within a given panel.**

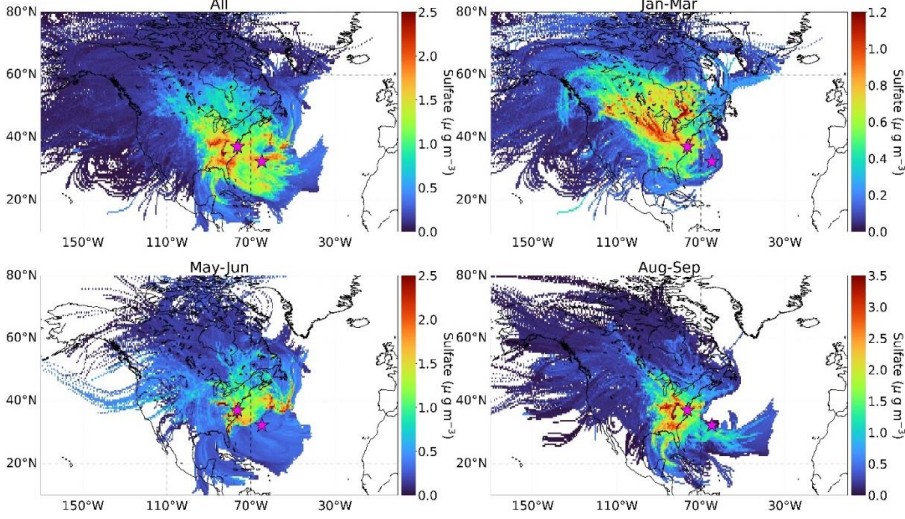

246

**Figure 3. Concentration weighted trajectory maps for sulfate aerosol concentrations as measured by an AMS on the Falcon during different ACTIVATE deployments (All data, Jan-Mar 2020 and 2021, May-Jun 2021, August-September 2020). These are based on 29,164 cloud-free AMS data points. The pink stars represent NASA Langley Research Center**





**(Hampton, Virginia) and Bermuda for reference. Color bar scales differ to show variability**
**better within a given panel.**

## 3.2 Droplet Residual Composition

A striking result in all seasons is that organic mass fraction was higher downstream of the
CVI in droplet residual particles in contrast to adjacent BCB and ACT legs in cloudy ensembles
(Figure 4). To compensate, sulfate mass fractions decreased in droplet residuals. Furthermore, $f_{44}$
increased in droplet residuals as compared to BCB and ACT data in each season, indicative of
more contribution of oxygenated organic species like carboxylic acids. There was no significant
difference in the mass fraction profiles between seasons for a fixed leg type (Figure 4).
The organic mass fraction and $f_{44}$ changes in droplet residuals can be explained by some
combination of preferential activation of CCN with these special properties and/or aqueous
processing in droplets to generate oxygenated organics. Although not the focus here, the high
chloride mass fractions in droplet residuals (Figure 4) can be explained by how sea salt would
preferentially activate into drops owing to its large size and that the AMS has some ability (albeit
not efficient) to detect sea salt chloride (Zorn et al., 2008; Ovadnevaite et al., 2012). These results
are important in that the usage of more readily available datasets such as MERRA-2 for speciated
aerosol data fail to capture the chemical characteristics of droplets contributing to $N_d$ (Section S1
and Figure S1), which are shown here to be distinctly different than what was measured below and
above cloud.

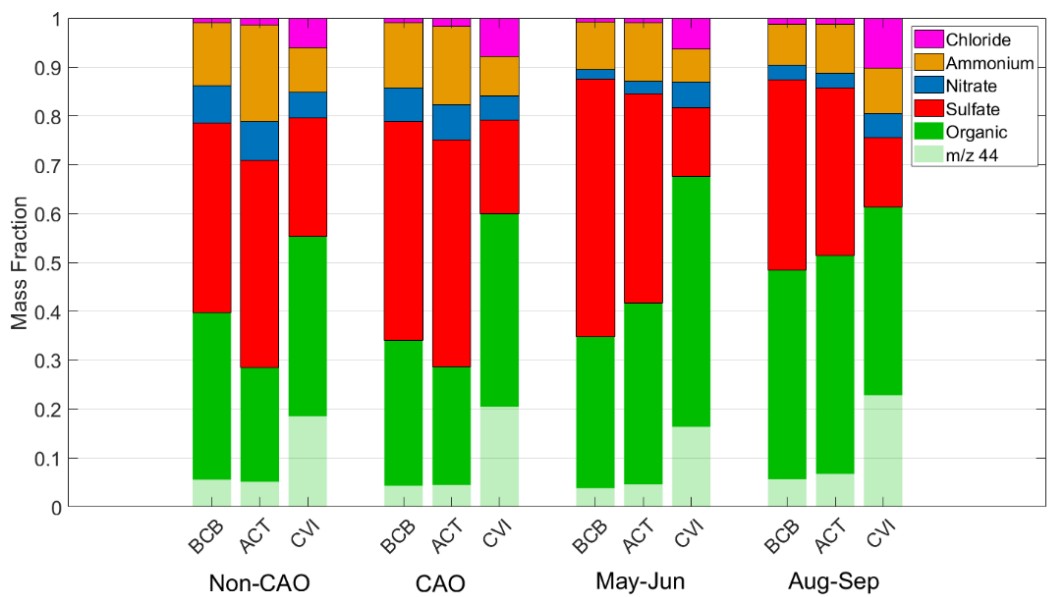


**Figure 4. Seasonal comparison of AMS mass fractions, including the relative contribution of**
**m/z 44 to total organic ($f_{44}$). Note that the Non-CAO and CAO categories represent all flight**





**data in January-March (deployments 1 and 3) that were separated using the criteria in**
**Section 2.4.**

277         We next examine scatterplots of organic mass fraction (i.e., organic mass divided by total
AMS mass) differences between each cloud leg with CVI-AMS data and its closest BCB leg in
the same cloud ensemble versus analogous sulfate mass fraction differences for the same pair of
legs (Figure 5). Aqueous processing to preferentially increase one of the two species relative to
the other would presumably translate into a positive value on the more preferred species' axis; in
other words, if there was more organic aerosol formation in clouds via aqueous processing
relative to sulfate, it would register as a positive (negative) value on the y (x) axis. Regardless of
season, the results reveal a consistent feature of increasing (decreasing) organic (sulfate) mass
fraction downstream of the CVI relative to BCB samples, suggestive of aqueous processing
shifting the composition to be more organic-rich. For the very few points laying to the bottom
left of the origin, nitrate is often more enhanced in those droplet residual samples relative to
BCB data.

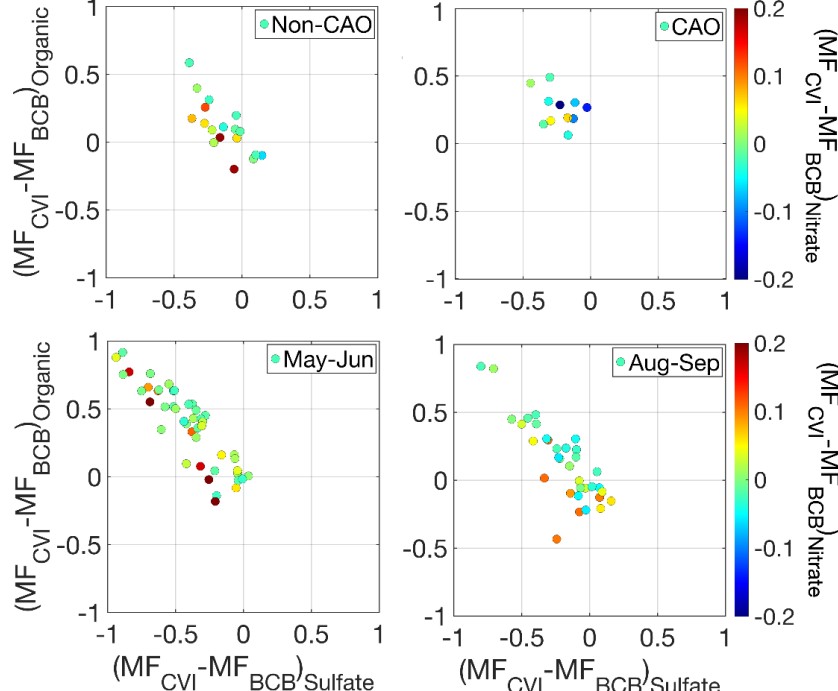


**Figure 5. Scatterplot of the difference in organic mass fraction in cloud legs with CVI data**
**and below cloud base (BCB) legs for an individual cloud ensemble relative to the analogous**
**difference for sulfate mass fraction between the same pair of legs. Markers are colored by**
**the analagous difference in nitrate mass fraction. Panels represent different seasons with**



**winter deployments (January-March) separated into cold air outbreak (CAO) and non-CAO**
**days.**

296          A comparison of $f_{44}$ versus $f_{43}$ in "triangle plot" format (Ng et al., 2010) shows an important
difference between CVI data and either BCB or ACT data in each season (Figure 6). Ambient
organic aerosol typically converge at the top left of the triangle representative of more atmospheric
aging leading to low volatility oxygenated organic aerosol species. The CVI data are
systematically higher and to the left of the triangle plot in each season. In contrast, the BCB and
ACT data are lower and to the right of the triangle plots without much distinction, suggestive of a
similarly lower level of oxygenation relative to droplet residuals.

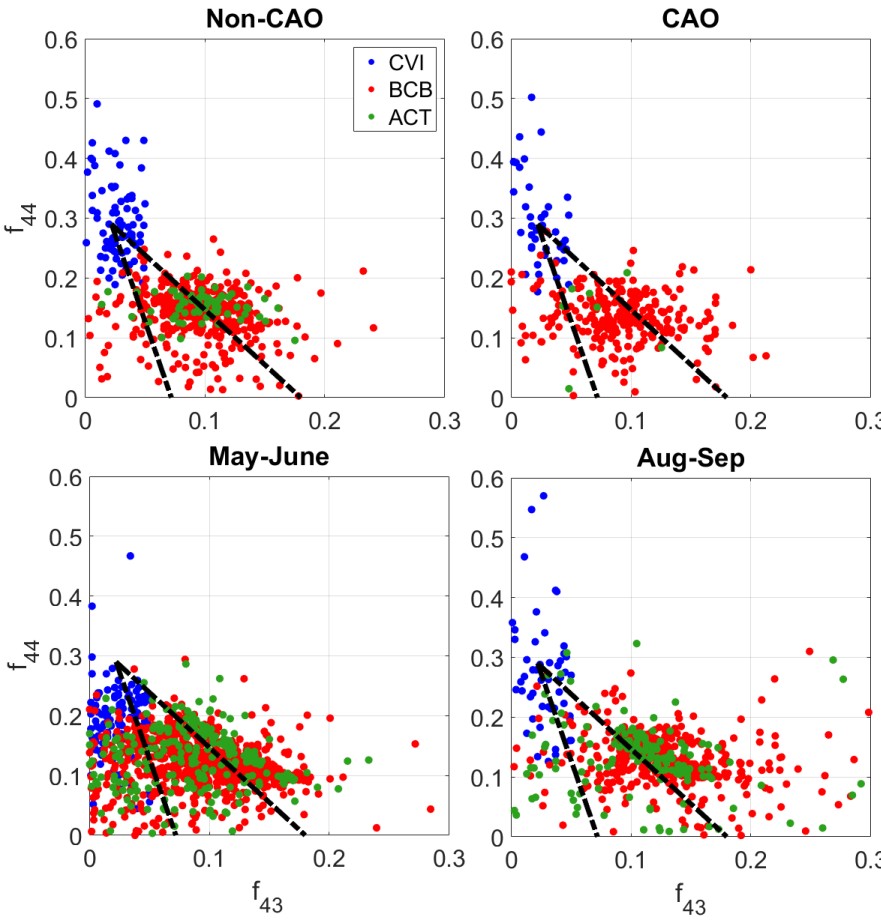


**Figure 6. Comparison of $f_{44}$ and $f_{43}$ for individual BCB and ACT legs out of cloud, in addition**
**to CVI data in cloud legs. Data are separated between time periods coinciding with different**
**ACTIVATE deployments. Superimposed on the plots are triangles corresponding to how**
**former work (Ng et al., 2010) compared these ratios. Points with organic mass concentration**
**less than 0.5 µg m$^{-3}$ were omitted from this analysis.**



The CVI droplet residuals are more oxidized because of some combination of aqueous
processing effects to yield more oxidized organic species, or because CCN with higher $f_{44}$ activated
into droplets. To probe more into which of the two aforementioned processes leads to the cluster
of CVI points at the top left of the triangle plots, we next examine (analogous to Figure 5)
scatterplots of $f_{44,CVI} - f_{44,BCB}$ versus $f_{43,CVI} - f_{43,BCB}$, where data are compared between the pair of
cloud and BCB legs closest to one another in individual cloud ensembles (Figure 7). If there was
no difference in organic composition between a pair of legs, a marker representing that pair would
be at the origin. Aqueous processing is presumed to result in a positive (negative) value on the y
(x) axis. Each season consistently exhibits points positioned to the top left of the origin suggestive
of aqueous processing leading to the enhanced oxygenation of the organic fraction in droplet
residuals relative to BCB legs. Note that this analysis omitted consideration of ACT legs as the
predominant source of droplets is from activation of sub-cloud aerosol particles.

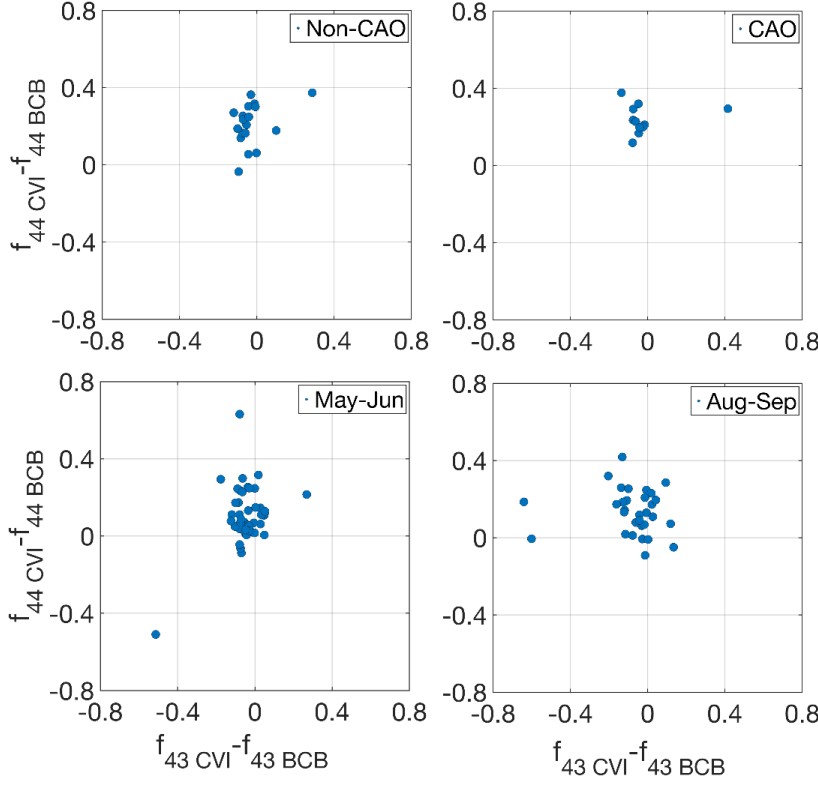


**Figure 7. Scatterplot of the difference in $f_{44}$ in cloud legs with CVI data and below cloud base (BCB) legs for an individual cloud ensemble relative to the analagous difference for $f_{43}$. Panels represent different seasons with winter deployments (January-March) separated into cold air outbreak (CAO) and non-CAO days.**






A brief discussion on possible artifacts is warranted including processes occurring in the
CVI inlet. First, we note that 23% of BCB/CVI pairs of data points (25 out of 110) exhibited higher
organic mass fraction in the BCB leg relative to droplet residuals (Figure S6), demonstrating that
the null case exists without an organic enhancement downstream of the CVI. The CVI inlet was
designed with both stainless steel and aluminum yielding negligible organic contamination
(Shingler et al., 2012). Also, the heated counterflow in the CVI reduces positive artifacts from
volatile gaseous species partitioning into sampled droplets such as with volatile organic
compounds (VOCs) to form organics or with nitric acid to form nitrate (Prabhakar et al., 2014); in
contrast, the heated counterflow would presumably evaporate some fraction of the existing nitrate
and organics in the CCN that activated into droplets unlike sulfate which is not volatile. Inlets
including the CVI can be prone to droplet shatter such as with large drizzle drops (> 100 µm)
(Twohy et al., 2013), although drizzle was not always frequent and the particulate artifacts
generated would still be representative of droplet residuals. It seems implausible that such drop
shatter would lead to an organic enrichment especially as this is observed across the entire study
region.
It is unclear why neither the BCB or ACT legs exhibit a composition profile matching the
droplet residuals since ultimately the droplet residual particles will evaporate outside of cloud and
return to the aerosol phase. Although difficult to prove with this dataset, a plausible explanation is
that the BCB and ACT particles have the added influence of interstitial particles in clouds that did
not activate into droplets. More research is needed to determine how repeatable such results are
for other regions, with simultaneous measurements of interstitial particles helpful to understand
why the droplet residual chemistry deviates from both the BCB and ACT data.

### 3.3 Cold Air Outbreak Case Studies

Owing to interest in the winter season having the strongest aerosol-cloud interactions
(Dadashazar et al., 2021a; Painemal et al., 2021), here we examine case study research flights
(RFs) during CAOs. Six CAO case study flights are used to understand the compositional
characteristics below, inside, and above clouds. Two flights are profiled here and the other four
are shown in Figures S7 (RFs 5-6 on 22 February 2020) and S8 (RFs 10-11 on 28 February 2020).
A representative day was 8 March 2020, which included two consecutive flights (RFs 17
and 18) based out of Hampton, Virginia profiling aerosol and cloud properties in CAO conditions.
These flights were investigated in past work showing enhanced new particle formation in ACT
legs (Corral et al., 2022) and that entrainment of free tropospheric air dilutes MBL CCN
concentrations (Tornow et al., 2022). Figure 8 shows the AMS composition profile on the out-
and-back flights, which involved flying out to a point and repeating the same path back to the
airfield. Stacked on top of each other in Figure 8 are the corresponding legs within individual cloud
ensembles including (from top to bottom) ACT, either BCT or ACB legs with CVI data, and BCB.
RF17 in the morning comprised 13 different cloud legs with corresponding BCB and ACT legs.
The BCB and ACT mass fraction profiles were similar with sulfate being most abundant (mass
fractions: 0.34-0.65) followed closely by organics (mass fractions: 0.15-0.42). The $f_{44}$ fraction of



the organics in BCB and ACT legs was quite low (0.00-0.16). The cloud data show a very different
profile with organics dominating the mass profile (mass fractions: 0.41-0.86) followed usually by
sulfate (mass fractions: 0.00-0.30). Furthermore, there was a significant jump in $f_{44}$ in the CVI data
(0.21-0.48). RF18 later in the day re-traced the same flight path and included 10 sets of matching
cloud-BCB/ACT legs showing again a similar jump in both organic mass fraction and $f_{44}$ in droplet
residuals. In the second flight there was more variability in the BCB and ACT pairs, with higher
sulfate mass fractions (0.34-0.75) in the ACT legs throughout most of the flight excluding the last
two sets of legs. The total AMS mass concentrations were slightly higher in the BCB legs (0.49-
0.91 μg m$^{-3}$) relative to ACT legs (0.24-0.50 μg m$^{-3}$). The other four flights shown in Figures S7-
S8 exhibit the same general results as those shown for 8 March with higher organic mass fractions
and $f_{44}$ in the cloud legs.

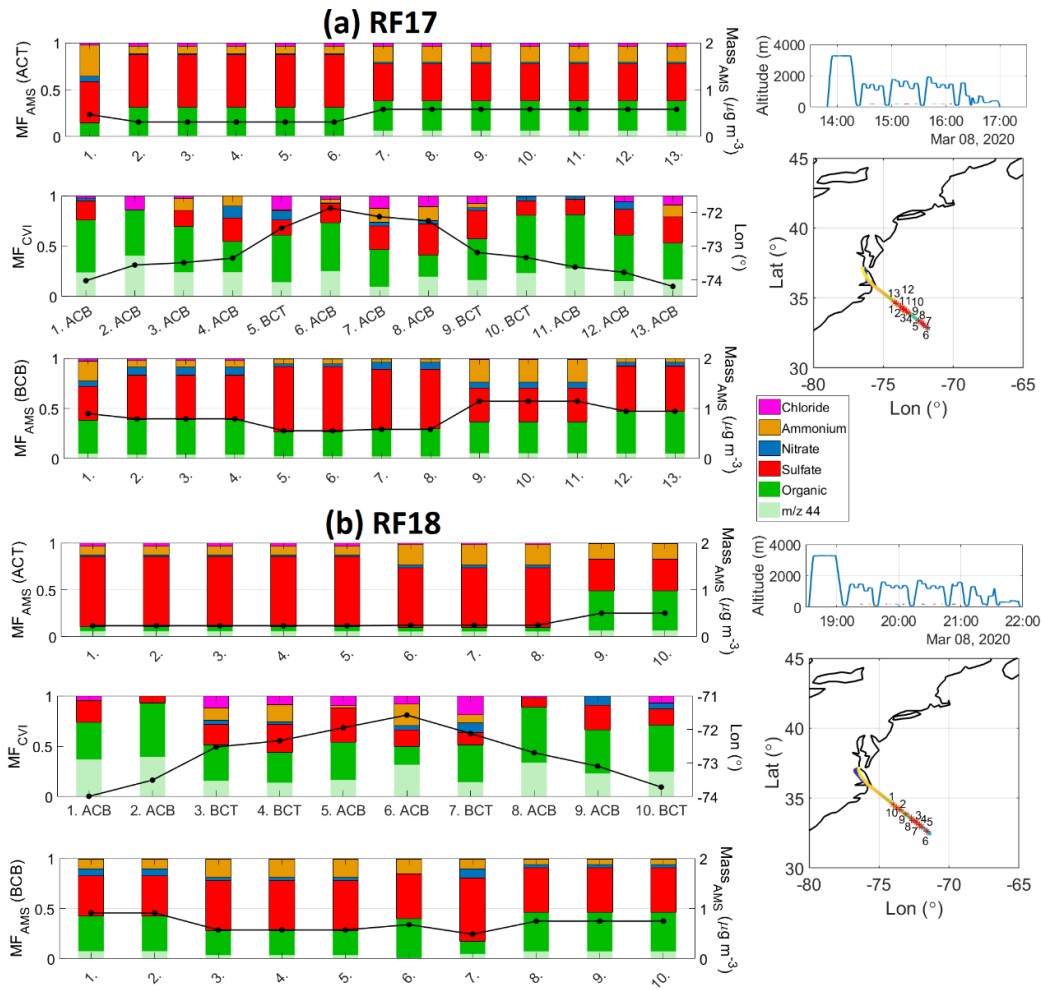


**Figure 8. Summary of AMS composition in adjacent BCB, cloud, and ACT legs during back-**
**to-back flights (Research Flights 17 and 18) in cold air outbreak conditions on 8 March 2020.**



**Shown in the bar charts are the mass fractions of AMS components in addition to either total AMS mass (for ACT and BCB legs; such data are not robust for CVI legs due to how the CVI operates) or longitude on the right y-axis. Note that some BCB and ACT legs are repeated for different cloud legs as they represent the closest leg to an individual cloud leg. On the far right are flight altitude during the flight along with the spatial map with numbers corresponding to the leg set numbers in the bar charts.**

## 4. Discussion

Our results represent unique atmospheric data that are scarce in the literature owing to the difficulty of obtaining aerosol chemical data below, in, and above cloud in close spatiotemporal proximity across many flights in different times of the year. Section S1 provides implications of the results in terms of differences with MERRA-2 speciated AOD. Although we cannot unambiguously prove it with the dataset, the results suggest processes in cloud changed the composition rather than preferential activation of CCN with enhanced values of the organic:sulfate ratio and $f_{44}$. That the droplet residuals shift to a more organic-rich signature with more oxygenated organics has implications for the aerosol particle properties remaining after droplet evaporation as they shift in composition and size. Having more organics relative to sulfate may reduce hygroscopicity at high RHs (e.g., Hersey et al., 2009), but a compensating factor could be that the organics are more oxygenated, which would increase the hygroscopicity of the organic fraction itself.

While a measurement of hygroscopicity of the droplet residuals was not available, Figure 9 shows an inverse relationship between f(RH) and organic mass fraction across all the BCB legs in ACTIVATE deployments 1-4, which is similar to what has been observed over the continental U.S. (Shingler et al., 2016); using the linear best fit line shows that the representative f(RH) value for pure organic aerosol (i.e., organic mass fraction of 1.0) was 1.22 in contrast with 0.92 over the continental United States (Shingler et al., 2016). The f(RH) value for pure inorganic aerosol (i.e., organic mass fraction of 0.0) was 1.39. Results of Figure 9 along with previous discussion suggests that aerosol interaction with clouds decreases particle hygroscopicity at an RH of 80% although future work will look deeper into aerosol hygroscopic properties over the ACTIVATE region. This is especially relevant as regulatory activities have reduced sulfate levels over the eastern U.S. in recent decades promoting higher relative amounts of organics (Bates et al., 2005; Hand et al., 2012) with downwind impacts on the northwest Atlantic due to offshore flow (Keene et al., 2014; Aldhaif et al., 2021; Dadashazar et al., 2021b).

Past studies provide a consistent story backing up the findings of this work. Coggon et al. (2012) showed increased AMS organic:sulfate ratios with altitude in the marine boundary layer over the northeast Pacific Ocean coincident with increased liquid water content, which was attributed to aqueous processing effects to generate more organics relative to sulfate; this was also suggested by past work in that region with a particle-into-liquid sampler (Sorooshian et al., 2007). Coggon et al. (2012) showed that organics and sulfate were typically the most abundant AMS species both below cloud and in droplet residuals with comparable mass fractions and no consistent trend of either one dominating the droplet residual composition. Past measurements off the California coast and over Texas revealed enhanced $f_{44}$ values in droplet residuals relative to below and above cloud data and also relative to interstitial aerosol particles in cloud (Sorooshian et al., 2010). That study showed similarly enhanced values of other ratios in droplet residuals indicative



of more oxygenated organics (e.g., PILS oxalate:AMS m/z 44, PILS oxalate:AMS organic). Over
the Texas area, PILS measurements of oxalate relative to AMS sulfate and organic revealed
significant enhancements (factors up to 4 and 13, respectively) downstream a CVI relative to
cloud-free conditions at similar altitudes (Wonaschuetz et al., 2012); furthermore they showed that
organic mass fractions increased together with oxalate:organic and oxalate:sulfate ratios as a
function of residual cloud fraction, which was a metric representing "cloud processing history" of
an air parcel in shallow cumulus cloud fields. CVI-AMS data from a surface site studying warm
tropospheric clouds on Mt. Åreskutan in central Sweden in July 2003 showed that organics and
nitrate activated with higher ease than sulfate (Drewnick et al., 2007); even though our results
suggest the droplet residual changes in composition are largely driven by aqueous processing, it is
relevant that organics have been shown in at least another region to activate more easily than
sulfate.

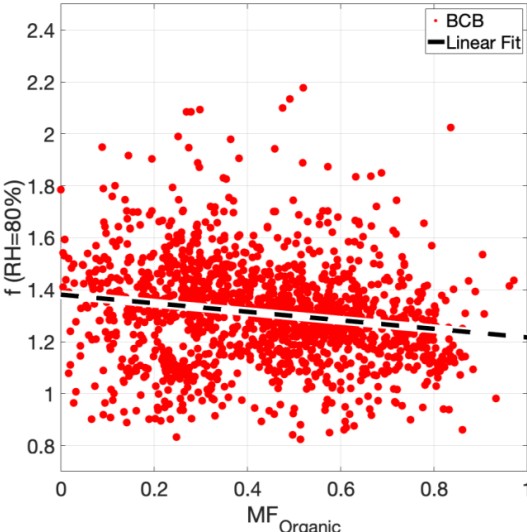


**Figure 9. Relationship between f(RH) and organic mass fraction for BCB legs during
ACTIVATE deployments 1-4. Markers are based on f(RH) data synched to the time
resolution of the AMS data. The f(RH) values from the linear fit at a MF$_{organic}$ value of 0.0
1.0 are 1.39 and 1.22, respectively.**


## 5. Conclusion

A large airborne dataset collected over the northwest Atlantic as part of the NASA
ACTIVATE mission show a distinctly different chemical signature in cloud droplet residuals
(lower sulfate mass fraction, higher organic mass fraction, and higher f$_{44}$) relative to particles
below and above cloud. Detailed analysis suggests this shift in composition is driven more by in-
cloud aqueous processing rather than preferential activation of CCN with such chemical
characteristics. Several case study flights during cold air outbreak conditions are profiled showing
the aforementioned compositional changes in droplet residuals. More work is needed to both



validate whether aqueous processing is the primary reason for the composition changes and to
determine if these results apply to other regions.

453       The results of this study are critical in motivating increased attention to both in-cloud
formation of oxygenated organics and the composition of particles activating into droplets over
the northwest Atlantic. Furthermore, this work advances knowledge of aerosol-cloud interactions
in this region as datasets often relied on in the absence of airborne data such as reanalysis data
suggest a different story where sulfate is more enhanced than organics year-round (in contrast to
the airborne data) (e.g., Braun et al., 2021). Cloud processing is a source for organics that cannot
be ignored, especially in light of the increasing relative amount of species in aerosol particles other
than sulfate due to regulatory activities over the U.S. (Hand et al., 2012).


*Data Availability.*

ACTIVATE Airborne Data:
https://doi.org/10.5067/ASDC/ACTIVATE_Aerosol_AircraftInSitu_Falcon_Data_1
(NASA/LARC/SD/ASDC, 2020a),
https://doi.org/10.5067/ASDC/ACTIVATE_Cloud_AircraftInSitu_Falcon_Data_1
(NASA/LARC/SD/ASDC, 2020b), and
https://doi.org/10.5067/ASDC/ACTIVATE_MetNav_AircraftInSitu_Falcon_Data_1
(NASA/LARC/SD/ASDC, 2020c).

*Author contributions.* HD conducted the analysis. AS and HD prepared the manuscript. All authors contributed by providing input and/or participating in airborne data collection.

*Competing interests.* The authors declare that they have no conflict of interest.

*Acknowledgments.* The work was funded by NASA grant 80NSSC19K0442 in support of ACTIVATE, a NASA Earth Venture Suborbital-3 (EVS-3) investigation funded by NASA's Earth Science Division and managed through the Earth System Science Pathfinder Program Office. CV and SK thank funding by the DFG CRC 301 TP Change and by HGF W2W3-060. We acknowledge use of imagery from the NASA Worldview application (https://worldview.earthdata.nasa.gov/), part of the NASA Earth Observing System Data and Information System. We thank pilots and aircraft maintenance personnel of NASA Langley Research Services Directorate for successful execution of ACTIVATE flights.

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
