# Peer review of "Organic enrichment in droplet residual particles relative to out of cloud over the northwest"

_Atmospheric Chemistry and Physics, 2022_

## Author Comment (AC1)

Response: We thank the reviewers for reviewing our manuscript and providing feedback. Below we provide responses to comments and suggestions in blue font.

Reviewer 1:

This manuscript presents an interesting data set derived from airborne measurement by an AMS during the first 4 ACTIVATE campaigns over the Atlantic Ocean east of Virginia. The AMS sampled aerosol above and below marine boundary layer clouds, in and above the boundary layer in cloud free air; and cloud droplet residual particles sampled through a CVI. The main finding was that the droplet residual particles generally showed large enhancements of the organic mass fraction compared to aerosol sampled out of cloud, both very near to clouds as well as in clear air regions. The organic material in the droplet residuals also tended to be more oxidized than the organic material in aerosol out of cloud.

The authors make a convincing argument that the increased organic fraction in the droplet residuals is more likely the result of aqueous processing in cloud (their hypothesis) (or possibly in the CVI (my tentative alternate suggestion)) than preferential activation of the organic rich aerosol when the clouds formed. They suggest that incorporation, and apparent oxidation, of organic mass in cloud droplets will result in aerosol that is enriched in organics after the droplets evaporate, which could have important implications regarding the subsequent ability of the MBL aerosol population to act as CCN (organic rich aerosol might be less effective CCN than sulfate dominated aerosol). Most models do not include aqueous processes that enrich cloud droplets in organic matter, and tend to predict that sulfate should be the dominant component of the MBL aerosol and constitute the main fraction of the CCN population.

However, there is a critical aspect of the data set that makes it unclear whether the apparent enrichment of oxidized organic matter in droplet residual particles actually has a significant impact on the composition of MBL aerosol. One would expect that aerosol just above or below clouds that have droplets greatly enriched in organics would also show some enhancement compared to aerosol that had not been cloud processed. This was not observed. In fact, Table 1 shows that the average organic mass fraction of aerosol in cloud free BL was higher than observed just below clouds in all four of the bins (non-CAO, CAO, May-Jun, and Aug-Sep). The same is true when one compares the average organic mass fraction between clear air aerosol above the top of the boundary layer and that sampled just above clouds. Given that clear air sampling was "generally closer to the coast" (line 110) and thus generally upwind of the regions where clouds were sampled, these observations seem to suggest that cloud processing tends to increase sulfate relative to organics in the aerosol, regardless of the very large enhancement of organic mass fraction in the cloud droplet residual particles.

Response: These are all excellent points. We try to respond to each issue in the comments below and with revised text in the new draft. Note that as most of the cloud-free data was closer to land, it could just be that there are greater organic levels in the outflow relative to farther offshore, where sulfate presumably becomes more important due to marine emissions of precursors such as dimethylsulfide. The region's synoptic flow is not always strictly offshore from west-to-east. Thus, the higher organic content near the coast often could just be due to local emissions that are confined to the coast and aren't advected any farther east. Therefore, it is not fair in our view to use that argument that the higher organics in cloud free air by land are a sign of cloud

processing yielding more sulfate. Even if that was the case, one cannot also rule out that organics decreased simply due to dilution and removal processes whereas sulfate went up with distance offshore owing to marine sources.

A question that requires thinking though is that if the source of the enhanced production of organics is in droplets, then shouldn't the strongest signal be observed in that reservoir rather than out of cloud? This is what past works have found too. We had posed this hypothesis is in our original draft and still retain the following text:

"Of particular interest related to aerosol chemical characterization around clouds is growing evidence in the literature that in-cloud aqueous processing can generate not only sulfate (Barth et al., 2000; Ervens, 2015) but also secondary organic aerosol (SOA) (Blando and Turpin, 2000; Warneck, 2003; Sorooshian et al., 2006a; Ervens et al., 2011; Heald et al., 2011), which is hypothesized to manifest itself in enhanced organic mass fractions in droplet residuals relative to below and above cloud."

The authors do admit this "problem" nearly in passing (lines 342-344) but do not fully explore or explain it. Here I offer a number of possibilities that should be considered. First, there may be some artifact in the CVI causing the enrichment in oxidized organics measured by the AMS. This does not seem too likely but needs to be investigated and ruled out. Another possibility could be that the cloud drops truly are greatly enriched in organics, but when the droplets evaporate much of this material is also volatile enough to return to the gas phase without producing aerosol highly enriched in organics. Perhaps a large fraction of the organic rich cloud droplets continue to grow large enough to precipitate, again leaving no highly enriched aerosol behind. My final possible explanation may be the most optimistic. Maybe there is evidence of organic-rich cloud-processed aerosol in the data set that is lost in the averages but clear in case studies. For example, might the increase in organic mass fraction ACT in cloud legs 5 and 6 on RF 5 (Fig S7) be a hint of evaporated cloud droplets vented above the clouds (of course it is a little problematic that the organic mass fraction remains high ACT for the rest of the flight). It might be worth looking particularly at BBL and ABL legs downwind of, and soon after, cloud legs to see if there are organic rich cloud processed aerosol there (possibly contributing to the high averages reported in Table 1).

Response: As we said in our response above, we find the results of the study to still be robust and that aqueous processing still very well could be taking place. However, this is an excellent comment that the paper would benefit from more discussion about other possible factors at play.

We added a figure in the Supplement and placed more attention on the issue of artifacts in the inlet, which we still do not believe explains the results. We talk more about this in response to another comment farther below and show the new Figure S3.

For the argument about organics volatilizing, our argument to refute that possibility is that the CVI heats incoming droplets and evaporates them in the inlet already, which should have removed those vulnerable species already prior to droplet evaporation outside the inlet. This would be a very simple explanation though if the CVI didn't already heat and evaporate droplets.

There is complexity with trying to study any relationships of our results with precipitation since it is too difficult to claim causal effects; going down that path would just complicate the story and be more distracting than helpful in our experience. We note that we did compare the results to both rain water content and ice water content (ice was prevalent for a few of the winter flights) and did not see any relationships pointing to a distinct effect of precipitation on our results.

In terms of there being a signal out of cloud being lost due to the averaging, that is a valid point as there can be variability in the AMS composition during level legs. The new Figure S3 we show of a time series during a research flight demonstrates this variability along different level legs. There is variability at fine scales that is not consistent with the organic:sulfate ratio always being enhanced on the CVI, but of course statistically the overall results show that that trend is dominant. As you can tell already in the responses, this is not a straightforward matter as the processes at play are difficult to disentangle and there is much that takes place in the ambient atmosphere that is difficult to replicate in a lab as it relates to characterizing inlet performance.

In summary, as noted at the start, this is an interesting data set, but I am not convinced that the suggested implications are supported. The inference that organic rich particles reaching the AMS behind the CVI are reflecting the composition of cloud droplets and will lead to organic rich aerosol needs to be better supported.

Response: Thank you for the thoughtful perspectives on the study's interpretations. You make excellent points that required additional discussion which we have now included. We open by stating that although it may not have been emphasized well in the last draft, the results of an enhancement of a chemical parameter behind a CVI that is not as distinct out of cloud is not new. Others have shown a similar type of phenomenon for other regions using other instrumentation (e.g., Sorooshian et al., 2010; Coggon et a., 2012; Wonaschuetz et al., 2012) – we emphasize this more throughout the text but especially in Section 4. It certainly is interesting that the results are the way they are and naturally leads one to be skeptical and associate it with instrument artifacts or other factors unrelated to cloud processing. But the fact that the majority of cases show the strongest signal behind the CVI is not alarming to us as we have confidence in the measurements and data, and because the signature of aqueous processing presumably would be high in the actual reservoir where it is taking place rather than out of cloud. As we note in the text now:

"The previous discussion does not provide support for any form of artifact or contamination explaining why 74% of the CVI data points exhibited higher organic mass fractions than both the BCB or ACT legs. One could argue that the chemical signature of cloud processing should be evident out of cloud somewhere as ultimately the droplet residual particles will evaporate outside of cloud and return to the aerosol phase. As will be discussed in Section 4 though, there is a body of literature pointing to droplet residuals having the strongest signature of cloud processing rather than below or above cloud. Although difficult to prove with this dataset, a plausible explanation is that the processed aerosol dilutes into the MBL at a time-scale that is much faster than the production/evaporation cycle."

We view this paper (as were the other ones cited above) as stepping stones to continue studying these types of data to learn more about droplet residual composition and more broadly cloud processing. There is so little data documented for cloud droplet residuals that we believe this paper is valuable, even if there is still uncertainty as to why a clear signature was not observed out of cloud in every cloud case we sampled. A key aspect of this research is that aerosol-cloud interactions are extremely complicated. We present here a high quality dataset that we have QC'd tremendously and stand firmly behind. We hope this reviewer will find our revisions and responses acceptable.

A number of editorial comments follow, keyed to line numbers:

62 does higher or lower MBL height and liquid cloud fraction "promote stronger aerosolcloud interactions"?

Response: Fixed by revising the text

"…higher marine boundary layer height, higher low-level liquid cloud fraction…"

68/69 "Or particular" should be "Of particular"

Response: Fixed

76-79 Not clear to me how knowing the composition of droplet residuals directly tells us about composition of CCN, especially if there is aqueous processing active in the droplets.

Response: What was partially meant here is that even if there was aqueous processing, this would affect the droplet residual particle (after drop evaporation) participating in a future activation process. We tried to clarify with this text:

"…chemical analysis of droplet residuals can potentially lend insight into properties of the CCN activating into droplets from the current or a future cloud passage…"

108 It seems like the cloud-free ensembles ought to be closer to 15 minutes than 35.

Response: You are correct. The clear ensembles were approximately 15 min long (~13 min and 37 seconds on average). We clarify this in the text:

"Cloud-free ensembles were approximately 15 min (~100 km)."

128-131 Rationale for ignoring cloud water composition could be simplified to just note that the partial speciation of organics in the cloud water samples makes it hard to compare to AMS total organics.

Response: Fixed

"Note that while cloud water samples were also chemically characterized, those data are outside the scope of this work as the partial speciation of organics in the cloud water samples makes it hard to compare to AMS total organics."

132-133 Might be prudent to just say that smearing in the PILS used in ACTIVATE was problematic, and not imply that all PILS suffer extreme smearing. See Zeng et al., 2021 (AMT, 14, 6357-6378, https://doi.org/10.5194/amt-14-6357-2021 for a description of how modifications greatly reduced smearing in the PILS NOAA flew during FIREX AQ.

Response: Sure, this is simpler to say. Text now reads:

"…and chemical smearing during sample collection…"

136-142 Why were trajectories only calculated for cloud free points (I assume this means BBL and ABL points)? If you actually have trajectories for BCB and ACT legs as well this should be made clear.

Response: We originally did use data from BCB and ACT legs. Here is the revised text around those lines of text and also in the Figure S3 caption:

"Note that this includes data during cloud ensembles but only when cloud liquid water content was < 0.05 g m$^{-3}$, and thus data during BCB and ACT legs are included."

215 seems to be missing one or more words. Maybe something like "source aloft that might include..."

Response: Thank you for catching this and we changed it to now say "source aloft that might include..."

217 should "boundary clouds" be "boundary layer clouds"?

Response: Yes it should. It is now fixed.

229-230 double check Table 1. I only looked closely at Organic and Sulfate but noticed that for ACT Non-CAO the concentrations show Org>SO4 but MF has SO4>Org. Same is true for ACT May-Jun and also for ABL CAO.

Response: We doublechecked the numbers to confirm that indeed there was no error originally. We realize it may seem counter-intuitive, but it can happen that average organic mass is higher than sulfate but the average mass fraction of sulfate is higher than organic. That's because we are averaging mass fractions over all the measurements and we are not calculating average mass fraction from grand average of masses. We add here (not in paper, but just in this response file) a figure showing a time series (x-axis represents different ACT legs) of mass and mass fraction overlaid by their averages for the example of non-CAO ACT legs to show that the calculations were correct.

[Figure]

266-270 related to comment on lines 76-79. If your main conclusion is that aqueous processing is adding a lot of organics to cloud droplets, it is not certain that measuring the composition of droplet residual particles tells us anything definite about the composition of the CCN, which are what the Nd is dependent on.

Response: We removed the old lines 266-270 in question as they were not needed anyhow. Lines 76-79 are still valid since droplet residual composition "can potentially lend insight into properties of the CCN activating into droplets" – even if we don't show this unambiguously this possibility should still exist in principle. We feel no text changes are needed around the old lines 76-79.

288-289 Why include nitrate in Fig 5, seems pretty distant from your story?

Response: We included nitrate as we thought it would be good to lend some insight about why points were sometimes to the bottom and left of the origin in Figure 5 as noted in the text below. Since we do not think it hurts to leave it, we want to suggest we leave it the way it was.

"For the very few points laying to the bottom left of the origin, nitrate is often more enhanced in those droplet residual samples relative to BCB data."

319-320 Probably should have justified the exclusion of ACT legs in discussion of Fig 5 before this.

Response: We had done the analysis for ACT legs too and just didn't show it. We wrote in the following text near the end of discussion about Figure 5:

"Although not shown, the results in Figure 5 are similar to if ACT data were used in place of BCB data."

327-341 Not compelling evidence that there are no CVI artifacts.

Response: We added a new Figure S3 which shows a time series of a representative flight with quick passes in and out of cloud when the CVI was on. The data show that there is insufficient contamination to skew the data results of the study. We add text about the new Figure S3 to the section discussing possible artifacts and how there is also considerable variability in organics, not just behind the CVI but also behind the isokinetic inlet. We have given this extensive thought and stratifying the data by other parameters such as prevalence of rain/ice or by the shape of the overall droplet size distribution will not help with clearly stating any clear conclusions about artifacts versus other phenomenon. The bottom line is that the material of construction should not give off organics and our blanks out of cloud throughout the campaign do not give us any concern about contamination that would affect results of this study. We go over all of this in the expanded Section 3.2.

[Figure]

**Figure S3. Time series from research flight 10 (28 February 2020) of (top) AMS mass concentrations of sulfate and organic in addition to $f_{44}$, and (bottom) Falcon altitude and FCDP droplet number concentration. The CVI Flag horizontal bar indicators represent when aerosol sampling was conducted downstream of the CVI, with the vertical orange bars indicating specifically those CVI times where sampling was out of cloud (i.e., data serving as blanks).**

342-348 As noted in my review, this is a big problem that needs much more careful discussion than these few sentences. Seems you are suggesting that the interstitial aerosol (that would not appear in the droplet residuals) may be very enriched in SO4, masking any organic-rich cloud-processed aerosol coming out of the cloud. But the BBL and ABL samples suggest that aerosol in airmasses upwind of the clouds tends to be organic-rich, especially compared to BCB and ACT.

Response: We removed that sentence about interstitial aerosol as that seemed to be distracting and maybe not a strong idea. We disagree that this is a big problem as this has been observed in other regions and with other instrumentation in published works. We feel this finding is important to document for the course of this line of research into droplet residual composition with airborne measurements, which has a major scarcity of information in the literature.

As we explained earlier, the comparison of clear air ensemble data (ABL/BBL) to the cloud ensemble data may not be terribly fair owing to the closer proximity of the former legs to land where the aerosol is typically organic rich as shown by at least Corral et al. (2021; https://doi.org/10.1029/2020JD032592) who analyzed surface IMPROVE data along the East Coast of the U.S. Synoptic flow patterns vary and are not always offshore from west-to-east. Thus, the higher organic content near the coast may just be due to local emissions that are confined to the coast and aren't advected any farther east. We have extended the text in Sections 3 and 4 though to give more attention to these types of concerns though.

401-413 Discussion of Fig 9 is interesting, but you have not established that any of the organics in aerosol sampled outside of cloud were added to the aerosol phase through aqueous processing in the cloud.

Response: We still believe the old Figure 9 (now Figure 12) is still useful to show in light of how most of the existing cases with higher organic mass fraction out of cloud relative to in cloud are in the below cloud base leg. We precede the discussion of that figure with the following new text:

"While a measurement of hygroscopicity of the droplet residuals was not available, we instead examine aerosol hygroscopicity from BCB legs as that is the area out of cloud most commonly exhibiting higher organic mass fractions relative to in cloud. Even if the signature out of cloud is not as clear as one would expect presumably owing to dilution effects, still the influence of cloud processing on organics inevitably should exist to some extent making the subsequent discussion valuable."

443-460 Conclusions probably need to be revisited if you cannot establish that the droplet residual composition ever truly contributes significant aerosol with enhanced organics after the clouds dissipate.

Response: We have revised the conclusions but a key point here is that this study is not alone in terms of showing maximum values of compositional values in CVI data relative to either below or above cloud (e.g., Sorooshian et al., 2010; Coggon et al., 2012; Wonaschuetz et al., 2012). That being said, still in this dataset we present at least a quarter of the cases do still show enhanced organic mass fraction out of cloud.

Reviewer 2:

This manuscript summarizes three periods of measurements aboard the HU-25 Falcon over the western Atlantic of aerosol composition below, within and above clouds. It is an interesting data

set. The main conclusion is that the residual particle composition from the in-cloud samples is enriched in oxidized organic compounds 75% of the time. It is an interesting story but is not told well graphically.

1. Table 1 is very hard to read and compare. Put this in the supplement and tell the story graphically in the text.

Response: We have done this. See the new revised Figure 4 that summarizes almost all of the information in the old Table 1 (which is now Table S1):

[Figure]

**Figure 4. Seasonal comparison of AMS mass fractions, including the relative contribution of m/z 44 to total organic ($f_{44}$). Numbers above each bar represent the mean total AMS mass concentration for that category; note that absolute masses are not reported downstream of a CVI owing to high uncertainties. Note that the Non-CAO and CAO categories represent all flight data in January-March (deployments 1 and 3) that were separated using the criteria in Section 2.4.**

2. Figure S5 is important to the story and should be in the text.

Response: Done.

3. Figure S6 is the heart of the story and should be in the text.

Response: Done.

4. Figure 1 shows the flight locations and range of values but is not useful for comparing between the plots in the figure. I don't see any point in Figure S2.

Response: We removed Figure S2 and revised one sentence in the article file to now simply just say the following without any reference to a figure:

"Clear ensembles were generally closer to the coast."

We were unclear if the reviewer was requesting for us to modify or remove Figure 1. We feel it is fine to leave it in to give readers a sense of where data were collected and to compare values between species and seasons.

5. Figures 2, 3, S3, and S4 show trajectories but these data are never used in the paper. I don't see any point in including them.

Response: We feel that in these types of studies having the context of air mass history can be helpful even if they are not utilized extensively in the analysis and discussion. As a compromise, we removed Figures S3-S4 and moved the original Figures 2-3 into the Supplement (now Figures S1-S2). They still are useful for our discussion and referred to in the text.

6. Figure 4. Isn't m/z 44 included in "organics". If it isn't part of the mass fraction it shouldn't be included in this bar graph. The same for S7 and S8.

Response: An important aspect of the results is how "$f_{44}$" (ratio of m/z 44 to total organic mass) varies between the categories we are comparing. On these various bars it should be obvious to viewers that the m/z 44 contribution to the organic bars varies and that is an important aspect of our results. Therefore, we believe it is important to retain the current visual display of the relative contribution of m/z 44 to total organic mass.

7. Figures S7 and S8 should be in the text.

Response: We moved those figures into the article file.

8. Is the comparison with MERRA-2 part of this story? I think it is and therefore S1 should be in the main manuscript.

Response: We moved the MERRA-2 comparison to the draft, including Figure S1 (which is now Figure 1).

References:

Coggon, M. M., Sorooshian, A., Wang, Z., Metcalf, A. R., Frossard, A. A., Lin, J. J., Craven, J. S., Nenes, A., Jonsson, H. H., Russell, L. M., Flagan, R. C., and Seinfeld, J. H.: Ship impacts on the marine atmosphere: insights into the contribution of shipping emissions to the properties of marine aerosol and clouds, Atmos. Chem. Phys., 12, 8439-8458, 10.5194/acp-12-8439-2012, 2012.

Sorooshian, A., Murphy, S. M., Hersey, S., Bahreini, R., Jonsson, H., Flagan, R. C., and Seinfeld, J. H.: Constraining the contribution of organic acids and AMS m/z 44 to the organic aerosol budget: On the importance of meteorology, aerosol hygroscopicity, and region, Geophysical Research Letters, 37, https://doi.org/10.1029/2010GL044951, 2010.

Wonaschuetz, A., Sorooshian, A., Ervens, B., Chuang, P. Y., Feingold, G., Murphy, S. M., de Gouw, J., Warneke, C., and Jonsson, H. H.: Aerosol and gas re-distribution by shallow cumulus clouds: An investigation using airborne measurements, Journal of Geophysical Research: Atmospheres, 117, https://doi.org/10.1029/2012JD018089, 2012.

---

## Author Response (AR2)

Response: We thank the reviewers for reviewing our manuscript and providing feedback. Below we provide responses to comments and suggestions in blue font.

Reviewer 1: Accepted as is
Response: Thanks for the support.

Reviewer 2:
I appreciate the consideration the authors gave to comments from Reviewer 2 and myself, and changes they made in response. In particular I am pleased by the additional attention given to establishing that the observed enrichment of organics in residual particles is very unlikely to be a measurement artifact, and the expanded discussion of the lack of evidence for a clear impact of cloud processing on the composition of aerosol sampled just above or below clouds.

As noted in my first review, it seems misleading or ill-advised to suggest that measuring the composition of residual particles provides insight into the composition of the CCN that activated to form the cloud drop, especially when the main point of the paper is to suggest that aqueous processing in cloud drops significantly modifies the composition of the residual particles. I urge the authors to consider reworking 2 sentences in the introduction to address this concern. Specifically I would change the sentence in lines 84-87 to something like: "Furthermore chemical analysis of droplet residuals should lend insight into the properties of the aerosol that will be released after the droplet evaporates which could control its propensity to activate in a subsequent passage through cloud, with past work........(ref as they are)". Likewise, I would suggest deleting the last sentence of the introduction to remove this misleading assertion (any modification that I might suggest for this sentence would be largely redundant with those outlined for lines 84-87.)

Response: We modified Lines 84-87 as suggested: "Furthermore, chemical analysis of droplet residuals should lend insight into the properties of aerosol particles that will be released after droplets evaporate, which could control their propensity to activate in a subsequent passage through cloud, with past work showing an important role for organics (Russell et al., 2000; Drewnick et al., 2007; Mertes et al., 2007; Hawkins et al., 2008; Asa-Awuku et al., 2015)."

As suggested, we also deleted the last sentence of the introduction.

When I read the response to reviewers I was mildly alarmed by the assertion that "Others have shown a similar type of phenomenon for other regions....." (First paragraph in the third block of blue text/author response). Issue is that the Sorooshian group apparently made all of the other measurements in other regions using other techniques that are cited in this sentence, making this not truly independent confirmation. As a result I read section 4 quite closely, and found that the language describing prior work is much more careful in the revised text than in the response document.

Response: We interpreted this comment as saying that the manuscript text was written carefully and is thus adequate, whereas they felt the response file text was "mildly alarming". Either way, we are confident the manuscript text is fine as is in Section 4 in terms of addressing past work in relation to results of this study. As a result, we make no further changes for this comment.

Below are a small number of editorial suggestions to consider.

lines 145-146 For consistency it seems that the manufacturer of the 2DS-V probe should be provided.

Response: Done – added "(SPEC, Inc.)".

lines 236-238 Is this sentence needed? No attempt is made anywhere in the paper to explain why eastern North Atlantic should be so different than the western part focused on in ACTIVATE. Just pointing out such a major difference raises a lot of questions in some readers' minds that beg answers. If you are not going to try to give the answers why throw a possibly troublesome bit of trivia onto the table?

Response: That sentence is certainly not needed so we removed it.

line 245 is it important that chloride is low? Much later the fact that the AMS is not very sensitive to SS chloride is acknowledged; should that be mentioned here instead (if you do not choose to stop the sentence at the comma and delete any mention of Cl)?

Response: We just removed the part in question from that line: ""

lines363-375 I like Figure 4, but have to point out that this figure makes it much easier for the reader to see a point I raised last time; i.e the organic mass fraction is generally higher in cloud free air than it is in the residual particles sampled during the same season. I feel it would be a good idea to point this out and explain it (as done in the response) rather than hope readers do not notice.

Response: Sure, good point. We added the following text to that section: "The higher organic mass fractions in the BBL/ABL legs of clear ensembles relative to BCB/ACT legs of cloud ensembles can be explained by how most of the clear ensemble data were collected closer to land where there are greater organic levels in the continental outflow relative to farther offshore where sulfate presumably becomes more important due to marine emissions of precursors such as dimethylsulfide. The region's synoptic flow is not always strictly offshore from west-to-east. Thus, the higher organic content near the coast often could just be due to local emissions that are confined to the coast and are not advected any farther east."

lines 402-403 change to "Comparing CVI-AMS data to the closest ACT leg in the same ensemble gives similar trend (not shown)."

Response: Change made.

line 471 delete "with", or change "with conducting" to "to conduct"

Response: Change made: "A way to test this is to conduct CVI..."

line 617 Not sure you know that the aerosols resulting from cloud processing will shift in "size" compared to the precloud CCN. Very plausible that they will gain organic mass, so a dry aerosol might be larger, but if the extra organics make the particle less hygrosopic it would take up less water and might

be smaller.

Response: Change made to mention that there is a possible shift in size but not guaranteed: "That the droplet residuals shift to a more organic-rich signature with more oxygenated organics has implications for the aerosol particle properties remaining after droplet evaporation as they shift in composition and possibly size."

line 643 change "another" to "one other"

Response: Change made.